# Nerve Sheath Myxoma in Pregnancy: A Case Report

**DOI:** 10.3390/diseases12070164

**Published:** 2024-07-18

**Authors:** Elena De Chiara, Valerio Gaetano Vellone, Jacopo Ferro, Chiara Trambaiolo Antonelli, Liliana Piro, Stefano Avanzini, Valentina Prono, Andrea Beccaria, Monica Muraca, Ramona Tallone

**Affiliations:** 1Department of Neuroscience, Rehabilitation, Ophthalmology, Genetics and Maternal-Infantile Sciences, University of Genoa, 16126 Genoa, Italy; 2Pathology Unit, IRCCS Giannina Gaslini Institute, 16147 Genoa, Italy; valeriogaetanovellone@gaslini.org (V.G.V.); jacopoferro@gaslini.org (J.F.); chiaratrambaioloantonelli@gaslini.org (C.T.A.); 3Department of Pediatric Surgery, IRCCS Giannina Gaslini Institute, 16147 Genoa, Italy; lilianapiro@gaslini.org (L.P.); stefanoavanzini@gaslini.org (S.A.); 4Pediatric Radiology, IRCCS Giannina Gaslini Institute, 16147 Genoa, Italy; valentinaprono@gaslini.org; 5D.O.P.O. Clinic, Department of Pediatric Hematology and Oncology, IRCCS Giannina Gaslini Institute, 16147 Genoa, Italy; andreabeccaria@gaslini.org (A.B.); monicamuraca@gaslini.org (M.M.)

**Keywords:** nerve sheath myxoma, neurothekeoma, leg soft tissue tumor, immunohistochemistry, peripheral nerve sheath tumor, pregnancy

## Abstract

Nerve sheath myxoma (NSM) is a rare benign peripheral nerve sheath tumor that affects young adults. NSMs are asymptomatic, slow-growing swellings located in the upper extremities, more rarely in the lower extremities. Given the high risk of recurrence, it is recommended to perform a complete exeresis. To our knowledge, the evolution and management of NMS during pregnancy have not been described yet. We report the first case of recurrent pretibial NSM in a pregnant girl and its follow-up and outcome during and after pregnancy. NSM is difficult to diagnose clinically or using imaging. The final diagnosis remains histopathological. It is known how various types of benign and malignant skin tumors can develop or change during pregnancy. With our case, however, we documented that pregnancy does not affect the growth and evolution of NSM. Given the benign nature of the lesions and their tendency to grow slowly, during pregnancy, follow-up of NSMs can be conducted through ultrasonography and surgical treatment postponed after delivery. Our case highlights the importance of careful monitoring and individualized decision making, especially in rare scenarios such as NSM, where data on the progression of benign lesions are limited. Our case highlights the importance of a careful monitoring and a tailored treatment in rare scenarios such as NSM, where data on the progression of benign lesions are limited. Considering the benign nature of the lesions and their tendency to grow slowly, follow-up of NSMs during pregnancy can be conducted through ultrasonography, and surgical treatment can be postponed after delivery.

## 1. Introduction

NSMs are rare cutaneous neoplasms of peripheral nerve sheath origin with a strong predilection for the upper extremities, particularly in the hands and fingers, and are characterized as benign tumors that affect young adults. These tumors appear to affect males and females with equal frequency [1,2,3].

Although the terms neurothekeoma and nerve sheath myxomas were used interchangeably, they are now described as distinct entities, with the later exhibiting positive immunohistochemistry for S100 and Sox-10 and moderate/diffuse reactivity to glial fibrillary acidic protein (GFAP) and CD57 [4,5,6].

Treatment is primarily surgical, with clear margins, especially due to the high recurrence rates.

Given the rarity of the tumor, there are insufficient data in the literature to determine the most appropriate follow-up during pregnancy. This case describes the clinical, radiological–histopathological, and immunohistochemical features of recurrent NSM in a young pregnant girl and its management and outcome during the pregnancy.

## 2. Case Report

We present a case of a previously healthy 15-year-old Caucasian female that developed a superficial right pretibial mass (20 × 10 × 15 mm) characterized by slow and progressive millimetric growth.

In accordance with the literature, the girl underwent an excisional biopsy that revealed a histological pattern of neoplasm with a multilobular architecture within the dermis and partly in the subcutaneous tissue, expansile margins, and lobules containing abundant myxoid stroma compatible with a diagnosis of nerve sheath myxoma. The proliferation index (Mib1) was low, with a diffuse expression of S100. The excision was <1 mm from the surgical margin, and although the lesion had no malignant features, further enlargement was suggested, which the family preferred not to perform.

Two years later, a local clinical recurrence characterized by the appearance of two tense–elastic nodular lesions was observed. An ultrasonography showed four solid hypoechogenic nodules of which the largest was 6 × 2 mm. However, due to the benignity and the stability of the new lesions, it was decided, considering the family’s wishes, to place the girl in follow-up and not to proceed with further surgery. Clinical and radiological follow-up with ultrasound or magnetic resonance imaging (MRI), every six months, demonstrated a substantial stability for four years.

Within the oncological follow-up, at the age of 21, she presented two miscarriages (at 6 and 8 gestational weeks, respectively) in the absence of gynecological abnormalities. Her personal and family history was negative for thrombotic events except her sister who had congenital heart disease and polyabortion. For this reason, a level I and II coagulation screening and a karyotype analysis were performed, which were negative.

At the age of 22, new, denser, and bigger lesions were detected sonographically. The MRI described the presence of elongate/serpiginous nodules within the anteromedial subcutaneous tissue of the middle third of the right leg (maximum dimensions approximately 4 × 20 × 17 mm AP × LL × CC). Signal characteristics included homogeneous hyperintensity on STIR and T2, hypointensity on T1-weighted images, signal restriction on diffusion sequences, and discrete contrast enhancement after contrast medium administration. Additionally, a targeted radiological study excluded erosions along the cortical profile of the tibia (Figure 1).

Thus, given the developmental course of the lesions, it was decided that tumor removal would proceed, but shortly before surgery, the patient became pregnant. For this reason, in consideration of previous polyabortion and the report of a threatened abortion, the patient refused again to undergo the surgery, preferring to postpone it after the end of pregnancy.

Close ultrasound follow-up was then started at each trimester of pregnancy, which showed stability of the lesions. Gestation continued without further problems and ended with the birth of a healthy infant (with adequate parameters) via eutocic delivery. In addition, considering the previous history of recurrent abortions and the rarity of the tumor, placental analysis was also performed, which revealed the presence of multiple acute and chronic thrombotic events (Figure 2). No further coagulation tests were performed due to the well-being of the newborn.

Finally, the patient underwent surgery at the age of 23. The histological report documented macroscopically a dermal-hypodermal lesion with a polynodular architecture. Microscopically, elements appeared well defined, containing elements with polymorphic morphology, immersed in abundant myxoid stroma, partly fusiform, and epithelioid or stellate. An immunohistochemical analysis showed intense and diffuse expression of S100, SOX10, and glial fibrillary acidic protein (GFAP) molecules, with focal expression of cytokeratin AE1-AE3 and no expression of the MUC-4 molecule.

These nodules were surrounded by a thin fibrous capsule with very rare fusiform elements with the eosinophilic cytoplasm positive on immunohistochemical investigations for the epithelial membranous antigen (EMA). No clear mitotic figures (typical or atypical) were observed in the examined sections; the proliferative index (assessed by ki67) was about 10% (Figure 2).

The observed findings were consistent with the pattern of dermal nerve sheath myxoma with neoplasm-free margins. Three months later, the patient showed no clinical recurrence and will continue an ultrasound follow-up every six months.

## 3. Discussion

NSMs are rare benign peripheral tumors of the nerve sheaths, first described by Harkin and Reed in 1969 [5]. There is no prevalence between males and females, although there are authors who describe higher incidence in women [5]. The peak of incidence is in the fourth decade, even if pediatric presentation is described [6,7].

Patients usually present painless, solitary, superficial, multinodular masses that measure 0.5–2.5 cm. NSMs appears more frequently on the upper extremities (86% of cases), in particular the hand and fingers, and when it occurs in lower extremities, the knee/pretibial zone is the most common location [4].

Imaging studies do not have specific features through which to diagnose NSMs. Benign lesions usually tend to present a more homogeneous appearance. There are not many reports that illustrate the radiological characteristics of NSMs [8,9]. Our case, as reported by Salem and Tafti, shows typical features of hypoechogenicity with a posterior acoustic enhancement at US, and hyperchogenicity at T2-weighted MRI. Despite this, the final diagnosis remains histopathological [8,9].

In the past, NSMs and neurothekeomas (NTs) have been misdiagnosed and confused with each other because of their clinical appearance and similar behavior. The distinction from neurothekeoma has been clearly established since the article by Fetsch et al. which highlighted that careful examination of histology and immunohistochemistry reveals substantial differences between the two entities [4].

Nerve sheath myxomas typically manifest as superficial, highly myxoid, multinodular/multilobular masses with a peripheral fibrous border. These lesions consist of well-separated lobules of spindled, stellate-shaped, ring-shaped, and epithelioid Schwann cells, and immunohistochemical staining typically reveals reactivity for S-100 and Sox-10 proteins and GFAP, along with other neural markers, while showing limited reactivity for EMA and CD34. In contrast, NTs have characteristic epithelioid cells which are not immunoreactive to S-100 and Sox-10 proteins but are positive for vimentin, NKI-C3 (CD63), CD10, and Mit-F [4,6].

Treatment of NSM is surgical. The extent of surgical excision depends on factors such as tumor size, location, and histologic features and should aim to achieve negative margins. Although clear data on the margin needed to obtain radicality are not available in the literature, surgical treatment should aim for maximum resection, given the high rate of recurrence [4].

Local recurrence is possible in nearly 50% of cases, especially in the fingers, where surgery may be more challenging and difficult. Clear microscopic margins and grossly negative margins of a few millimeters are generally considered sufficient to minimize the risk of recurrence [9,10].

In our case, the patient, although there was indication, preferred, in accordance with the family’s requests, not to proceed with margin enlargement at first detection of NSM. This attitude, given the high probability of recurrence, resulted in the reappearance of lesions in the same area that had previously undergone surgery. The slow growth and benign nature of the lesion meant that the NMS remained stable for years. The subsequent growth of the lesions in number and size made it necessary to plan a new surgery, which the patient refused due to a high-risk pregnancy.

It is well documented how various types of benign and malignant skin tumors can develop or change during pregnancy, including melanoma, dermatofibroma, and pyogenic granuloma, likely due to the profound immunologic, endocrine, and metabolic changes that occur during pregnancy [9,11]. Other benign tumors of the peripheral nerve, such as neurofibromas, often first appear around the time of puberty, increase in number and size during pregnancy, and shrink after giving birth. McLaughlin et al. stated that the majority (75%) of neurofibromas expressed the progesterone receptor (PR), whereas only a minority (5%) of neurofibromas expressed the estrogen receptor [12]. Additionally, there are reported cases of schwannomas during pregnancy [13]. Regarding neurothekeoma, a benign nerve tumor similar to nerve sheath myxoma, De Giorgi et al. suggest a possible estrogenic influence. Estrogens impact cells via nuclear ERα and ERβ and membrane receptors like GPR30, activating MAPKs such as ERK, which regulate gene transcription and interact with signaling pathways. To date, the clinical significance of estrogen receptors in mesenchymal tumors remains uncertain; therefore, further investigations on a larger patient cohort are necessary for statistical validation [14].

There are no extensive studies in the literature on the evolution and course during pregnancy of both NSMs and NTs. The only two cases to our knowledge described in the literature concern a 31-year-old pregnant woman in her fifth month who had a myxoma of the nerve groin surgically removed and a 20-year-old woman with a rapidly growing neurothekeoma on the nose. Only in the last patient with NT did the authors conclude that cellular atypia and mitotic figures did not appear to correlate with aggressive behavior, but surgical treatment was deemed the primary choice of treatment [15,16].

In our case, however, we have documented that pregnancy and its associated hormonal and immunological changes do not seem to alter the growth and evolution of NMs. Follow-up, if clinical signs support the diagnosis, can be conducted through ultrasound imaging, and surgery can be postponed until after childbirth. However, this is a single case, making it difficult to draw definitive conclusions. Therefore, in cases of lesions suggestive of NSM, it is important to adopt a surgical approach, widening the resection margins if necessary, given the high recurrence rate.

Regarding our case, the presence of widespread recent and previous thrombi in the placenta remains to be clarified. The pregnancy was characterized by threats of miscarriage in the first trimester, but fortunately, it concluded with the birth of a healthy baby with appropriate length and weight for gestational age. The young woman, in fact, did not present alterations in the first- and second-level thrombophilic screening, including antiphospholipid antibodies, thrombotic events in her medical history or personal risk factors such as obesity or diabetes, maternal–fetal Rh or ABO incompatibility, preeclampsia/hypertension. The only relevant data remain the familial history, also present in her sister, of multiple abortions, for which unfortunately we lack documentation.

## 4. Conclusions

In conclusion, NSMs are rare benign skin tumors highly prone to recurrence if incompletely excised. We report the first case of follow-up and management and outcome of NSMs during pregnancy. The stability of the lesions during gestation demonstrates how the benign nature of these tumors might enable clinicians to postpone surgery after delivery, especially in cases where surgery can be a risk factor for the pregnancy itself. However, the high risk of recurrence underscores the importance of complete surgical excision when feasible. Our case highlights the importance of careful monitoring and individualized decision making, especially in rare scenarios such as NSM, where data on the progression of benign lesions are limited.

## Figures and Tables

**Figure 1 diseases-12-00164-f001:**
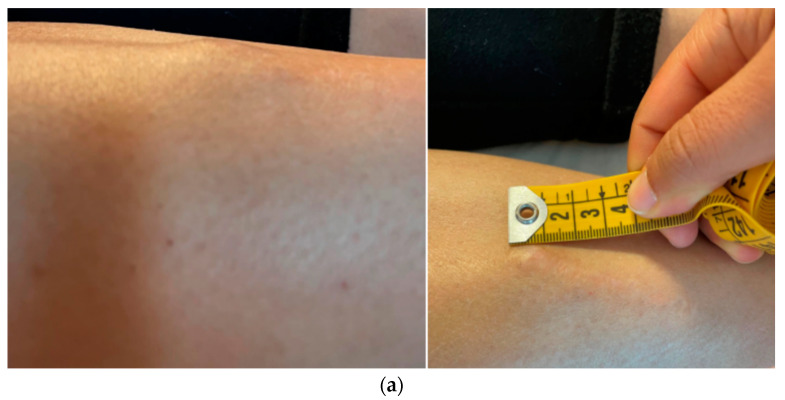
Macroscopic (**a**) and radiological (**b**,**c**) appearance of the lesion, hard nodulation of approximately 2 × 1.5 cm, underlying smaller “cluster” lesions, MRI: nodulations uniformly hyperintense in STIR, ultrasound: hypoechogenic lesions, with polylobate margins, partially confluent, without vascularization upon color Doppler analysis.

**Figure 2 diseases-12-00164-f002:**
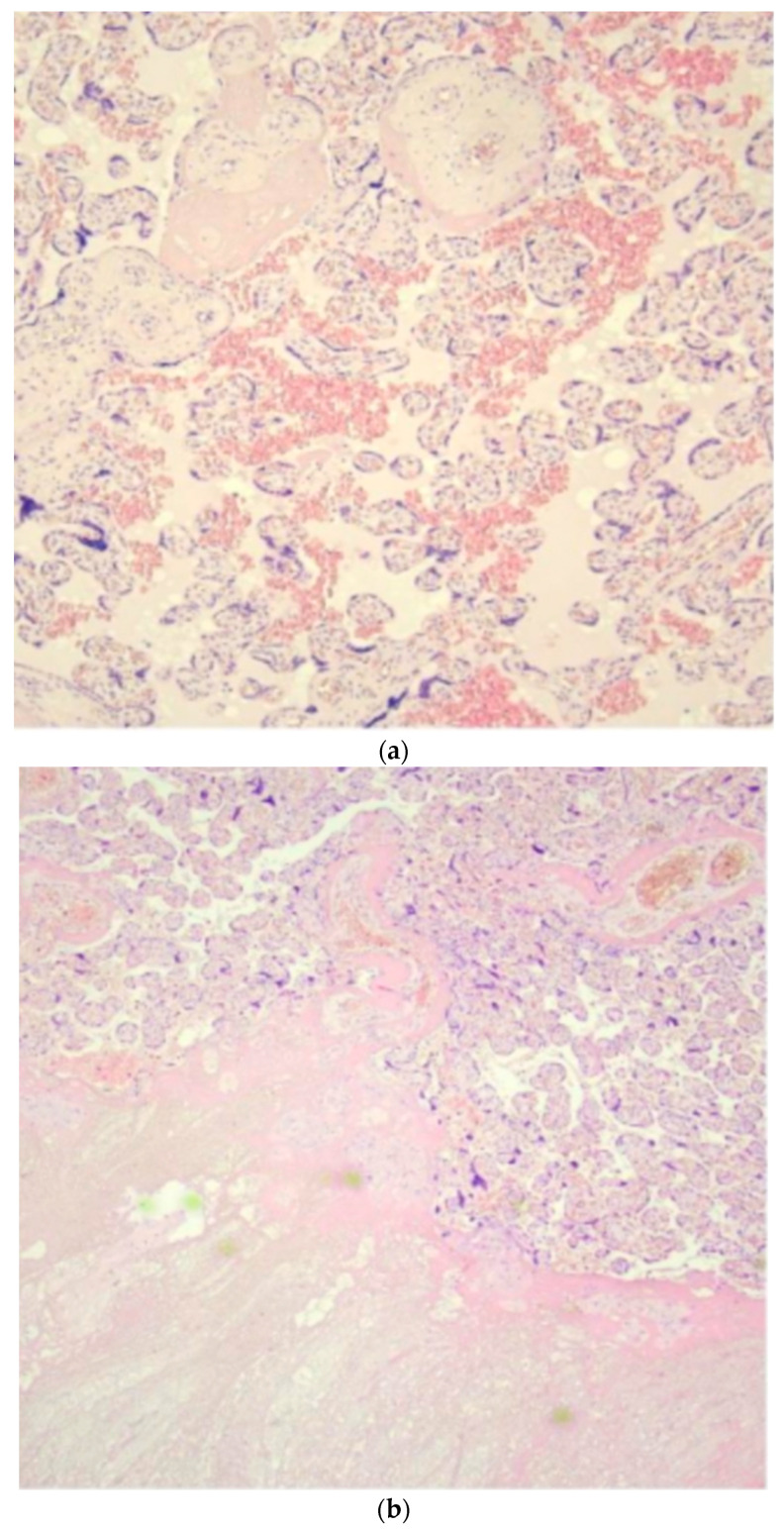
Histologic examination of placenta (**a**,**b**) and tumor (**c**–**e**): (**a**) long-standing hypoxic distress with recent and organized intervillous and retroplacental hemorrhages and recent and organized intervillous and retroplacental hemorrhages. (**b**) Paracellular ischemic lesions with coagulative necrosis. (**c**) 6×. Dermo-ipodermal lesion with polynodular growth and expansile borders. (**d**) 40×. Thin fibrous capsule surrounding each nodule. (**e**) 100×. Spindle and stellate elements in myxoid and edematous stroma.

## Data Availability

The original contributions presented in the study are included in the article, further inquiries can be directed to the corresponding authors.

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
