# Peer review of "Nerve Sheath Myxoma in Pregnancy: A Case Report"

_diseases, 2024, doi:10.3390/diseases12070164_

Round 1

Reviewer 1 Report

Comments and Suggestions for Authors

Dear authors, 

I read with interest your report on nerve sheath myxoma in pregnancy.

I found the report well written and potentially filling a current gap in the literature. 

In the discussion section, the authors could mention how skin diseases, particularly chronic inflammatory diseases such as psoriasis, have been reported to change during pregnancy (cite: DOI: 10.23736/S0392-0488.20.06748-6).

Comments on the Quality of English Language

Overall good quality

Author Response

Dear Reviewer,

Thank you for your comments. We have taken your excellent observation into account to expand the text as you suggested, highlighted in yellow. The literature indeed well documents the increased risk of melanocytic lesions, including melanoma. However, other chronic conditions such as atopic dermatitis, psoriasis, impetigo herpetiformis, or pemphigus may also appear or worsen. As suggested, we have included in the article how other skin diseases tend to change during pregnancy, and we have included the article you suggested in the references.

Reviewer 2 Report

Comments and Suggestions for Authors

The case study on the nerve sheath tumor in a pregnant patient indicates that pregnancy does not have a substantial impact on the development and progression of the tumor. Despite being uncommon, this type of tumor is understood to be unaffected by hormones. Therefore this result is expected.

While no other case studies are present in existing literature, drawing definitive conclusions remains challenging due to reliance on a single case.

The observed alterations in the placenta are relatively interesting, but there is no clear correlation with the tumor that is described.

In conclusion, I am not certain if this study offers enough novelty and interest to justify publication.

Author Response

Dear Reviewer,

Thank you for your comments.

The literature well documents the increased risk in pregnancy of melanocytic lesions, including melanoma. Other benign tumors of the peripheral nerve, such as neurofibromas, often first appear around the time of puberty, increase in number and size during pregnancy, and shrink after giving birth. McLaughlin et al. (McLaughlin ME, Jacks T. Progesterone receptor expression in neurofibromas. Cancer Res. 2003 Feb 15;63(4):752-5) stated that the majority (75%) of neurofibromas expressed the progesterone receptor (PR), whereas only a minority (5%) of neurofibromas expressed the estrogen receptor. Additionally, there are reported cases of schwannomas during pregnancy (Andrews S, Capen CV. Pelvic schwannoma in pregnancy. A case report. J Reprod Med. 1993 Oct;38(10):826-8).

Regarding the benign nerve tumor most similar to the nerve sheath myxoma, namely neurothekeoma, De Giorgi et al. hypothesize a possible hormonal influence by estrogen.de Giorgi V, Alfaioli B, Franchi A, Gori A, Sestini S, Papi F, Lotti T. Cellular neurothekeoma in a girl: could oestrogens favour the development and growth of this rare tumour? J Eur Acad Dermatol Venereol. 2008 Sep;22(9):1149-50. Estrogens affect cells through diverse mechanisms: the classical pathway via nuclear ERα and ERβ, and non-classical pathways involving membrane estrogen receptors like GPR30. These pathways activate MAPKs, such as ERK, which regulate gene transcription and interact with other signaling pathways. While the clinical relevance of estrogen receptors in mesenchymal tumors is uncertain, their findings suggest hormonal dependency in neurothekeoma cells, necessitating further study in a larger patient group for statistical analysis. For all the reasons stated, we believe that hormonal influence on NSM should not be excluded a priori.

Regarding the placenta, we authors were initially undecided about whether to include the data. However, after discussion, we decided to include it as additional information. Since there are no previous reports, it may be disconnected from the context, but further cases are needed to verify this.

Moreover, we acknowledge this as a case report. Nevertheless, nerve sheath myxoma is a rare tumor. Therefore, during our follow-up, it would have been beneficial to learn about other cases and their progression during pregnancy to validate our approach and provide reassurance to the patient. We believe that in cases of rare tumors or conditions, each individual case can contribute significantly to the scientific community, as every experience holds potential value. 

Reviewer 3 Report

Comments and Suggestions for Authors

The authors present a case of recurrent nerve sheath myxoma. submitted to excissional biopsy at 15 years of age,that presented local recurrence (without further surgery). When she was 22, surgery was yet proposed, but was not performed due to pregnancy, without significant modification during this time

It is a common experience that, despite the rarity, the rate of recurrence after incomplete excision was reported up to 47% (see i.e Fetsch JF et al. REF 4)

Therefore, despite the most intriguing point by the present report is related to the lack of impact on tumor growth by pregnancy, I think that the take home message is that a complete tumor removal must be proposed.

Author Response

Dear Reviewer,

Thank you for your insightful comments. We have reviewed the discussion and revised our article to emphasize the key message with which we fully agree. Initially, we chose to describe this case due to the uniqueness of the decisions made, given the patient's refusal to undergo surgery. However, we have updated the text to underscore the importance of complete tumor removal, considering the high recurrence rate.

Round 2

Reviewer 2 Report

Comments and Suggestions for Authors

Thank you for your answer for my review and your revision.

I still believe that the connection between nerve sheath myxoma and pregnancy/hormonal changes remains uncertain and very thin. The case report provides only limited evidence to support such connection.

As the authors note, while there is extensive literature on melanoma, much less information is available for most sarcomas.

However, given that the authors recognize the limitations of a case report in understanding the underlying mechanisms of this tumor, I agree to recommend this article for publication, if the editors believe it aligns with the journal's publication standards and mission.